# Association of Sarcopenia with Cognitive Function and Dementia Risk Score: A National Prospective Cohort Study

**DOI:** 10.3390/metabo13020245

**Published:** 2023-02-08

**Authors:** Ailing Lin, Ting Wang, Chenxi Li, Fan Pu, Zeinab Abdelrahman, Mengqi Jin, Zhenqing Yang, Liming Zhang, Xingqi Cao, Kaili Sun, Tongyao Hou, Zuyun Liu, Liying Chen, Zuobing Chen

**Affiliations:** 1Sir Run Run Shaw Hospital, Zhejiang University School of Medicine, Hangzhou 310016, China; 2The Second Affiliated Hospital and School of Public Health, The Key Laboratory of Intelligent Preventive Medicine of Zhejiang Province, Zhejiang University School of Medicine, Hangzhou 310058, China; 3Department of Neurobiology and Department of Orthopedics, The Second Affiliated Hospital, Zhejiang University School of Medicine, Hangzhou 310003, China; 4NHC and CAMS Key Laboratory of Medical Neurobiology, MOE Frontier Science Center for Brain Research and Brain–Machine Integration, School of Brain Science and Brain Medicine, Zhejiang University, Hangzhou 310003, China; 5Department of Rehabilitation Medicine, First Affiliated Hospital, College of Medicine, Zhejiang University, 79 Qingchun Rd., Hangzhou 310003, China; 6Center for Clinical Big Data and Analytics of the Second Affiliated Hospital, Department of Big Data in Health Science School of Public Health, The Key Laboratory of Intelligent Preventive Medicine of Zhejiang Province, Zhejiang University School of Medicine, 866 Yuhangtang Rd., Hangzhou 310058, China; 7Department of General Practice, Sir Run Run Shaw Hospital, School of Medicine, Zhejiang University, 3 Qingchun East Rd., Hangzhou 310016, China

**Keywords:** sarcopenia, cognitive function, BDRM score, older adult

## Abstract

The relationship between skeletal muscle and cognitive disorders has drawn increasing attention. This study aims to examine the associations of sarcopenia with cognitive function and dementia risk score. Data on 1978 participants (aged 65 years and older) from the 2011 wave of the China Health and Retirement Longitudinal Study, with four follow-up waves to 2018, were used. Cognitive function was assessed by four dimensions, with a lower score indicating lower cognitive function. Dementia risk was assessed by a risk score using the Rotterdam Study Basic Dementia Risk Model (BDRM), with a higher score indicating a greater risk. Sarcopenia was defined when low muscle mass plus low muscle strength or low physical performance were met. We used generalized estimating equations to examine the associations of sarcopenia. In the fully adjusted models, sarcopenia was significantly associated with lower cognitive function (standardized, β = −0.15; 95% CIs: −0.26, −0.04) and a higher BDRM score (standardized, β = 0.42; 95% CIs: 0.29, 0.55). Our findings may provide a new avenue for alleviating the burden of cognitive disorders by preventing sarcopenia.

## 1. Introduction

As one of the world’s top ten causes of death, dementia affects over 50 million people worldwide, and this number will triple by 2050 [1]. Mild cognitive impairment (MCI) is an intermediate phase between normal cognitive aging and dementia. An estimated 10 to 20% of older adults with MCI develop dementia over one year [2]. Up to now, no effective treatments for predominant dementia are available, though billions of dollars have been invested in this goal [3]. Therefore, preventing dementia and cognitive impairment is one of the general public health issues.

Sarcopenia, a common aging phenotype, has been defined as a progressive and generalized skeletal muscle disorder involving the accelerated loss of muscle mass and function [4]. Sarcopenia is associated with increased adverse outcomes, including falls, functional decline, frailty, and mortality [5]. A few cross-sectional studies have shown that sarcopenia might be a good indicator of poor cognitive function and dementia [6,7,8]. However, the association of sarcopenia with cognitive function and dementia risk has not been conclusive from longitudinal studies. A study among Japanese community-dwelling older adults found that sarcopenia was an independent risk factor for cognitive deterioration during the 1-year study period [9]. On the other hand, two cohort studies of French women and older Korean adults reported no significant association between sarcopenia and MCI [10,11]. More research is urgently needed, particularly in developing countries where the burden of cognitive disorders is dramatically increasing.

Therefore, this study aims to examine the associations of sarcopenia with cognitive function and dementia risk score in China, a developing country with the largest number of aging populations. 

## 2. Methods

### 2.1. Study Population

All participants were from the China Health and Retirement Longitudinal Study (CHARLS), a nationally representative sample of people aged 45 years and older [12]. The CHARLS began the national baseline survey in 2011 and included about 17700 participants in 150 counties/districts and 450 villages/resident communities. These participants were followed up every two or three years in 2013–2014, 2015–2016, and 2018-2019. The Biomedical Ethics Review Committee of Peking University approved the CHARLS, and all participants provided written informed consent. In this study, we first included 17,708 participants who attended the baseline survey of the CHARLS. After excluding participants whose ages were less than 65 years old, and with missing data on sarcopenia (n = 4735), cognitive function (n = 4012), and other covariates (n = 132), we included 1978 participants who were interviewed at least once in the follow-up waves. The detailed exclusion process is shown in Figure 1.

### 2.2. Assessment of Cognitive Function

The CHARLS assessed cognitive function in four dimensions: orientation; attention and calculation; episodic memory; and visuospatial ability. The individual measures are described in detail as follows: five questions were used to assess the participants’ orientation, the day of the date (year/month/day), the day of the week, and the current season. The participants’ orientation score was the aggregate number of the correct answers, ranging from 0 to 5. Participants were required to calculate serial subtraction 7 from 100 (up to five times) to assess their attention and calculation. The participants’ attention and calculation score was the aggregate number of correct answers, ranging from 0 to 5. The CHARLS used delayed recall to assess the participants’ episodic memory. Participants were required to recall ten Chinese nouns to the interviewer after answering about fifteen irrelevant questions. The participants’ episodic memory score was the aggregate number of the correct answers, ranging from 0 to 10. The participants were shown a picture of two pentagons overlapped and asked to draw a similar figure. Participants’ visuospatial ability was assigned a score of 1 if they “drew the picture” correctly; otherwise, it was assigned a score of 0. The global cognitive function score was the sum of orientation, attention and calculation, episodic memory, and visuospatial ability. The total score ranged from 0 to 21, with a higher score indicating better cognitive function.

### 2.3. Assessment of Dementia Risk Score

Because data on the diagnosis of dementia were not available in the CHARLS, we used the Rotterdam Study Basic Dementia Risk Model (BDRM) in this study to assess dementia risk score, which showed good discriminative ability in China [13]. The BDRM score was quantified by weighing dementia risk factors, including age, history of stroke, subjective memory decline, and need for assistance with finances or medications, with a higher score, indicating greater dementia risk (detailed information is shown in Appendix A).

### 2.4. Assessment of Sarcopenia

*Sarcopenia*: We adopted the recommended diagnostic algorithm of the Asian Working Group for Sarcopenia (AWGS) 2019 [14]. Sarcopenia was defined when low muscle mass plus low muscle strength or low physical performance were met.

Measurement of muscle mass: We adopted appendicular skeletal muscle mass (ASM) to estimate muscle mass [15], an anthropometric equation in the Chinese population:ASM = 0.193 × body weight + 0.107 × height − 4.157 × gender − 0.037 × age − 2.631

Body weight, height, and age were measured in kilograms, centimeters, and years, respectively. As for gender, men were given a value of 1 and women were given a value of 2. Based on the value of ASM, we calculated height-adjusted muscle mass (ASM/Ht^2^):Height-adjusted muscle mass = ASM/(height)^2^

Height was measured in meters. Following prior studies, the cut-off value for low muscle mass was the lowest 20th percentile of ASM/Ht^2^. In this analysis, participants were defined as having low muscle mass when their value of ASM/Ht^2^ was less than 5.28 kg/m^2^ for women and 7.01 kg/m^2^ for men.

Measurement of muscle strength: AWGS 2019 recommends handgrip strength to measure skeletal muscle strength [14]. In the CHARLS, participants were asked to try their best to squeeze the dynamometer at a right angle for a couple of seconds and then let go, alternating two measurements from the right and left hands. We used the average value to indicate the participants’ handgrip strength if they finished the measurement twice. If the participants were unable to perform grip strength measurement in one hand due to health reasons, we used the value measured by the other hand. Participants were defined as having low muscle strength when their value of handgrip strength was less than 18 kg for women, and 28 kg for men.

Measurement of physical performance: The CHARLS used gait speed for participants aged 60 years and older. For gait speed, participants were asked to walk 2.5 m at the usual pace twice. The average value was adopted in this study. According to AWGS 2019, participants were defined as having low physical performance when their walking speed was less than 1.0 m/s.

### 2.5. Covariates

Covariates included demographic factors (age, gender, marital status, education, residence), health status (Center for Epidemiologic Studies Depression Scale (CES-D) score, body mass index (BMI), disease count), and lifestyle (smoking status, alcohol consumption). Marital status was categorized as currently married or other. Education was categorized as having schooling experience (more than 1 year of education) or being illiterate. Residence was categorized as currently living in rural areas or a city/town. CES-D, as a screener for clinical depression, was used to measure the participants’ depressive symptoms. BMI was calculated as weight (in kilograms) divided by height (in meters) squared. Disease count included high blood pressure, diabetes mellitus, cancer, lung disease, heart problem, stroke, arthritis, liver disease, kidney disease, digestive disease, and asthma. All covariates have been reported to be related to cognitive function or dementia risk [16,17].

### 2.6. Statistical Analysis

Baseline characteristics of participants in total and in sarcopenia- and gender- specific subgroups were described. Continuous variables are presented as mean ± standard deviation (SD), and categorical variables are presented as absolute numbers and percentages. In this study, we used Student’s *t*-test to compare the differences between subgroups for continuous variables and Pearson’s chi-squared test for categorical variables.

Generalized estimating equations (GEEs) were used to examine the association of sarcopenia and its components with cognitive function and BDRM score, respectively. Cognitive function and BDRM score were standardized in the analysis. We adjusted for age, gender, and follow-up time in model 1 (because age was considered in the BDRM calculation, we did not adjust for age in any models for BDRM). We further adjusted for marital status, education, residence, CES-D, BMI, disease count, smoking status and alcohol consumption in model 2. We repeated the above analyses for three sarcopenia components, respectively. Coefficients and 95% confidence intervals (CIs) were documented.

To ensure the robustness of our findings, we further conducted a series of sensitivity analyses: (1) excluding participants with cognitive impairment at baseline; (2) adjusting for baseline cognitive function; and (3) excluding participants with extremely high BDRM score at baseline, i.e., more than mean +2SD. Given that the CHARLS has many middle-aged adults who are also susceptible to sarcopenia and cognitive disorders as reported in the literature, we reran our models in the total sample of the CHARLS. Furthermore, stratified analyses by gender were also conducted, given the gender difference in sarcopenia and cognitive disorders.

Statistical analyses were performed using Stata 16.0 (Stata Corporation). A two-sided *p* value < 0.05 was considered statistically significant.

## 3. Results

### 3.1. Baseline Characteristics of Study Population

Table 1 shows the baseline characteristics of the study participants. Participants with sarcopenia were more likely to be older, live in rural areas, have no schooling experience, suffer from more diseases (all *p* < 0.001), and have lower BMI and CES-D scores (*p* < 0.05). Participants with sarcopenia had significantly lower cognitive function and higher BDRM scores (all *p* < 0.001). Compared to males, females were more likely to live alone, have no schooling experience, have higher BMI, and have lower CES-D scores and lower cognitive function (all *p* < 0.001).

### 3.2. Association of Sarcopenia with Cognitive Function and Dementia Risk Score

Table 2 shows the association of sarcopenia and its components with cognitive function. In model 1, with the adjustment of age, gender, and follow-up time, compared to participants without sarcopenia, those with sarcopenia had significantly lower cognitive function (standardized, β = −0.38; 95% CIs: −0.51, −0.26, *p* < 0.001). In model 1, participants with low muscle mass (β = −0.34; 95% CIs: −0.44, −0.25, *p* < 0.001), low muscle strength (β = −0.38; 95% CIs: −0.47, −0.30, *p* < 0.001) or low physical performance (β = −0.37; 95% CIs: −0.56, −0.18, *p* < 0.001) had significantly lower cognitive function. After adjusting for more covariates, such as marital status, education, residence, CES-D, BMI, disease count, smoking status, and alcohol consumption, the above associations were maintained.

Table 3 shows the association of sarcopenia and its components with dementia risk score. In model 1, with the adjustment of gender and follow-up time, compared to participants without sarcopenia, those with sarcopenia had significantly higher BDRM scores (standardized, β = 0.55; 95% Cis: 0.42, 0.67, *p* < 0.001). In model 1, participants with low muscle mass (β = 0.38; 95% Cis: 0.29, 0.46, *p* < 0.001) and low muscle strength (β = 0.06; 95% Cis: 0.02, 0.10, *p* = 0.005) had significantly higher BDRM scores. After adjusting for more covariates, such as marital status, education, residence, CES-D, BMI, disease count, smoking status, and alcohol consumption, the above associations were maintained.

### 3.3. Sensitivity Analyses

In the sensitivity analyses, we did not observe significant changes regarding the association between sarcopenia and cognitive function when excluding participants with cognitive impairment at baseline or adjusting for baseline cognitive function (Appendix A). Similarly, we excluded participants with an extremely high (i.e., >mean + 2SD) BDRM score at baseline, and found that the association of sarcopenia with BDRM score was also maintained (Appendix A). When rerunning models in total sample of the CHARLS (N = 7759), the results were robust (Appendix A). Similarly, the association of sarcopenia with cognitive function and BDRM scores still remained in subgroups stratified by gender (Appendix A, e.g., for BDRM score in male: β = 0.46; 95% Cis: 0.28, 0.64, *p* < 0.001; in female: β = 0.38; 95% Cis: 0.19, 0.56, *p* < 0.001).

## 4. Discussion

Based on a sample of older adults from a nationally representative prospective cohort study, we found that sarcopenia was significantly associated with lower cognitive function and higher BDRM score. Further analyses suggested that low muscle mass, relative to the other two sarcopenia components, was more strongly associated with a higher BDRM score. Our findings highlight the importance of managing sarcopenia to prevent cognitive disorders.

Our results are consistent with previous studies conducted in the US, Brazil, and Mexico [8,16,18,19]. For instance, researchers have found that participants with sarcopenia at baseline were associated with higher risks of incident AD and MCI using MAP data [8]. One recent systematic review and meta-analysis evaluated the association between sarcopenia and MCI across 15 cross-sectional studies, including 10410 older adults from Asia, Europe, and America, and found a significant association between sarcopenia and MCI (Pooled OR = 2.25, 95% CIs = 1.70–2.97) [7]. However, one study of women who were 75 years or older in France found no significant association of sarcopenia with cognitive impairment [9]. Another one conducted in Korea on those aged 70-84 years found similar results [20]. The following several factors may explain the observed differences: (1) distinct measurements of sarcopenia and cognitive function/dementia; (2) different coverage of age groups; and (3) various follow-up periods. This study extended the findings to Chinese older adults, with a relatively large sample size and long follow-ups.

There are several explanations for the association between sarcopenia and cognitive disorders. First, sarcopenia and cognitive disorders share similar risk factors as follows: inflammation, characterized by interleukin-6 (IL-6) and tumor necrosis factor-α; oxidative stress; hormonal changes; and malnutrition. These factors might be critical for the two aging phenotypes [21,22,23]. Particularly, sarcopenia obesity may influence cognitive function largely due to malnutrition. Second, skeletal muscle could produce and secrete molecules, called myokines, which regulate brain functions such as learning and locomotor activity. Physical inactivity accompanied by sarcopenia also influences cognitive function by changing myokines’ circulating level [6]. Third, frailty allows sarcopenia to impose an effect on cognitive disorders [24,25]. Frail people exhibit deficits in executive control, especially the frontal cortex, raising the possibility of abnormalities in motor skills and executive mechanisms [26].

The stronger association of one sarcopenia component, namely muscle mass, with dementia risk score is another important finding in our study. This finding is consistent with that of some previous studies [27], whereas some other studies also highlight the importance of muscle strength [28]. More research is needed in diverse populations. The role of muscle mass in brain activity could be explained as follows. Skeletal muscle is a major source of neurotrophic factors, such as brain-derived neurotrophic factor, which regulates synapses. In addition, skeletal muscle activity is also important for modifying brain function through immune and redox effects [27]. Currently, a growing body of evidence suggested that nutrition and exercise interventions may ameliorate the severity of sarcopenia in older adults [29,30,31]. For example, a clinical trial investigated the effectiveness of Internet-based nutrition and exercise interventions in the elderly with sarcopenia. The results showed that skeletal muscle mass after intervention was higher in the nutrition plus exercise and nutrition groups than that in the control and exercise groups, which suggested that three months of high-quality protein intake can improve muscle mass in older adults with sarcopenia. [30]. These studies support the importance of improving muscle mass in older populations.

Our study has some strengths. To the best of our knowledge, this is the first study to explore the association of sarcopenia with cognitive function and dementia risk in China. We also examined the association for sarcopenia components. Second, we appreciate the relatively large sample size of the nationwide prospective cohort study, the broader age range, and the long follow-up period. The limitations of this study are as follows: First, calculating muscle mass using ASM was not as accurate as measuring muscle mass using professional instruments directly, which may bias our results. However, the CHARLS did not use bioelectrical impedance or X-ray to measure muscle mass. Second, the BDRM was not initially developed in China. However, it was the only choice given the unavailability of data on the diagnosis of dementia in the CHARLS and there are no alternative prediction models. Third, although we adjusted for potential confounders, residual confounding is still possible.

## 5. Conclusions

In summary, sarcopenia is significantly associated with lower cognitive function and higher dementia risk score in Chinese older adults. Our findings may provide a new avenue for alleviating the burden of cognitive disorders by preventing sarcopenia.

## Figures and Tables

**Figure 1 metabolites-13-00245-f001:**
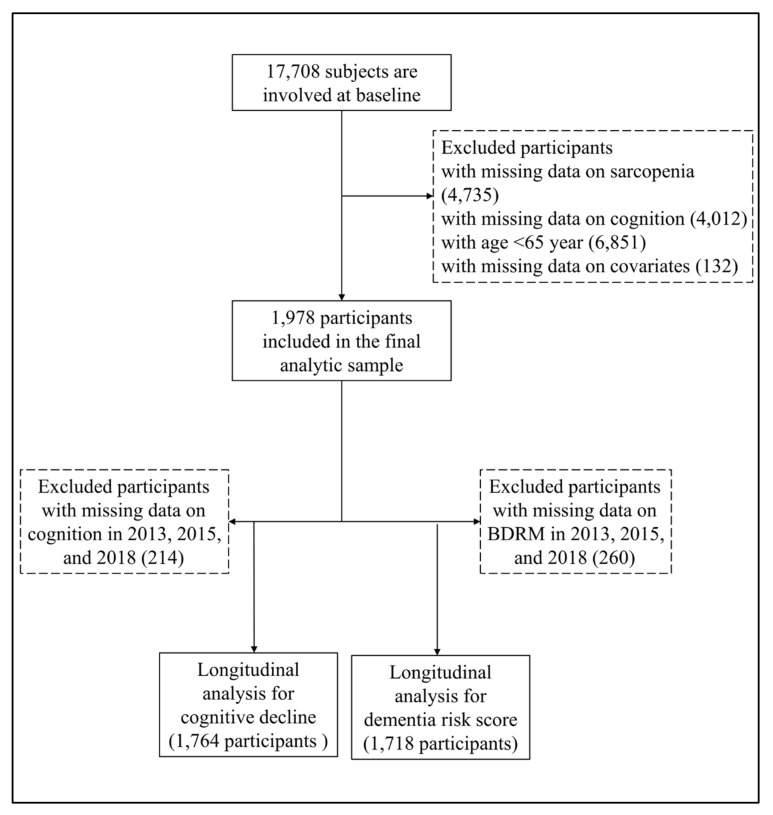
Flowchart for selecting study participants in this study.

**Table 1 metabolites-13-00245-t001:** Baseline characteristics of participants by sarcopenia and gender in CHARLS 2011.

	Sarcopenia			*p* Value ^a^		Gender	*p* Value ^a^
	Overall (N = 1978)	Yes (N = 256)	No (N = 1722)		Male (N = 1074)	Female (N = 904)	
Age, years	71.2 ± 5.3	74.1 ± 6.1	70.8 ± 5.0	<0.001	71.07 ± 5.0	71.34 ± 5.5	0.245
Female, N,	904 (45.7%)	119 (46.5%)	785 (45.6%)	0.788			
Rural, N	1469 (74.3%)	222 (86.7%)	1247 (72.4%)	<0.001	775 (72.1%)	649 (71.7%)	0.019
Currently married, N	1422 (71.9%)	151 (59.0%)	1271 (73.8%)	<0.001	876 (81.6%)	546 (60.3%)	<0.001
Illiterate education, N	710 (35.9%)	133 (52.0%)	577 (33.5%)	<0.001	218 (20.3)	492 (54.4%)	<0.001
Drinking, N	844 (42.7%)	107 (41.8%)	737 (42.8%)	0.762	690 (64.2%)	154 (17.0%)	<0.001
Smoking, N	918 (46.4%)	128 (50.0%)	790 (45.9%)	0.217	785 (73.1%)	133 (14.7%)	<0.001
CES-D score	16.6 ± 10.8	15.7 ± 10.4	16.7 ±10.9	0.040	17.4 ± 10.9	15.6 ± 10.7	<0.001
BMI, kg/m^2^	22.7 ± 4.1	19.4 ± 2.1	23.2 ±4.1	<0.001	22.3 ± 3.6	23.3 ± 4.5	<0.001
Disease count ^b^	1.6 (1.5, 1.7)	1.4 (1.3, 1.6)	1.6 (1.5, 1.7)	0.030	1.5 ±1.3	1.7 ± 1.3	0.029
Cognitive function	9.7 ± 4.0	7.6 ± 3.8	10.0 ± 4.0	<0.001	10.6 ± 3.8	8.5 ± 4.0	<0.001
Z-score	0.0 ± 1.0	-0.5 ± 0.5	0.08 ± 0.06	<0.001	0.2 ± 0.9	−0.3 ± 0.9	<0.001
Cognitive components							
Orientation	3.6 ±1.4	3.0 ± 1.3	3.7 ± 1.3	<0.001	3.8 ± 1.3	3.3 ± 1.4	<0.001
Memory	2.9 ±1.7	2.3 ± 1.5	3.0 ± 1.7	<0.001	3.1 ± 1.7	2.8 ± 1.6	<0.001
Attention and computation	2.6 ±2.0	1.9 ± 1.9	2.7 ± 2.0	<0.001	3.1 ± 1.9	2.0 ± 1.9	<0.001
Language	0.6 ± 0.5	0.4 ± 0.5	0.6 ± 0.5	<0.001	0.7 ± 0.5	0.4 ± 0.5	<0.001
BDRM score	78.2 ± 5.8	81.5 ± 6.7	77.7 ± 5.5	<0.001	78.0 ± 5.6	78.4 ±6.0	0.159
Z-score	0.0 ± 1.0	0.6 ± 0.5	−0.09 ± 0.07	<0.001	−0.03 ± 0.9	0.03 ± 1.0	0.159

SD, standard deviation; CES-D, Center of Epidemiological Studies—Depression; BMI, body mass index, BDRM: Rotterdam Study Basic Dementia Risk Model; CHARLS, China Health and Retirement Longitudinal Study. ^a^ *t*-test for continuous variables and a chi-squared test for categorical variables between participants with sarcopenia and without sarcopenia. ^b^ Diseases included high blood pressure, diabetes mellitus, cancer, lung disease, heart problem, stroke, arthritis, liver disease, kidney disease, digestive disease, and asthma. Continuous variables are presented as mean ± standard deviation (SD), and categorical variables are presented as absolute numbers and percentages.

**Table 2 metabolites-13-00245-t002:** Association of sarcopenia and its components with cognitive function.

	Model 1		Model 2	
	β (95% CI)	*p* Value	β (95% CI)	*p* Value
Sarcopenia				
No	Ref.		Ref.	
Yes	−0.38 (−0.51, −0.26)	<0.001	−0.15 (−0.26, −0.04)	0.005
Sarcopenia components				
Low muscle strength				
No	Ref.		Ref.	
Yes	−0.34 (−0.44, −0.25)	<0.001	−0.20 (−0.28, −0.12)	<0.001
Low muscle mass ^a^				
No	Ref.		Ref.	
Yes	−0.38 (−0.47, −0.30)	<0.001	−0.14 (−0.24, −0.05)	0.002
Low physical performance				
No	Ref.		Ref.	
Yes	−0.37 (−0.56, −0.18)	<0.001	−0.20 (−0.35, −0.04)	0.012

CI, confidence interval. Model 1 was adjusted for age, gender, and follow-up time. Model 2 was additionally adjusted for marital status, education, residence, smoking status, alcohol consumption, CES-D score, BMI, and disease count. ^a^ Because age was used to define low muscle mass, we did not adjust for age in models.

**Table 3 metabolites-13-00245-t003:** Association of sarcopenia and its components with BDRM score.

	Model 1		Model 2	
	β (95% CI)	*p* Value	β (95% CI)	*p* Value
Sarcopenia				
No	Ref		Ref	
Yes	0.55 (0.42, 0.67)	<0.001	0.42 (0.29, 0.55)	<0.001
Sarcopenia components				
Low muscle strength				
No	Ref		Ref	
Yes	0.06 (0.02, 0.10)	0.005	0.06 (0.01, 0.10)	0.011
Low muscle mass ^a^				
No	Ref		Ref	
Yes	0.38 (0.29, 0.46)	<0.001	0.40 (0.29, 0.52)	<0.001
Low physical performance				
No	Ref		Ref	
Yes	−0.03 (−0.10, 0.05)	0.534	−0.02 (−0.01, 0.07)	0.699

CI, confidence interval. Model 1 was adjusted for gender and follow-up time. Model 2 was additionally adjusted for marital status, education, residence, smoking status, alcohol consumption, CES-D score, BMI, and disease count. ^a^ Because age was used to define low muscle mass, we did not adjust for age in models.

## Data Availability

The data that support the findings of this study are available from the website of China Health and Retirement Longitudinal Study at
http://charls.pku.edu.cn/en.

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
