# Peer review of "Association of Sarcopenia with Cognitive Function and Dementia Risk Score: A National Prospective Cohort Study"

_metabolites, 2023, doi:10.3390/metabo13020245_

Round 1

Reviewer 1 Report

This paper aimed to examine the associations of sarcopenia with cognitive function and 30 dementia risk score..

It is well written and interesting and sounds to the publication after minor revision. 

minor comments

1)  table 1 should be stratified by gender, not only by sarcopenoa yes and no

2) it does not make sense not stratifying by gender. 

3) also table 2 must be stratified by gender

4) into the discussion please diffentiate the level of sarcopenia with the condition of sarcopenic obesity citing this paper. 

Perna, S., Peroni, G., Faliva, M.A., Bartolo, A., Naso, M., Miccono, A. and Rondanelli, M., 2017. Sarcopenia and sarcopenic obesity in comparison: prevalence, metabolic profile, and key differences. A cross-sectional study in Italian hospitalized elderly. Aging clinical and experimental research29(6), pp.1249-1258.

Author Response

Q1 table 1 should be stratified by gender, not only by sarcopenia yes and no. It does not make sense not stratifying by gender

Response: Thanks for your important notes. We have added basic characteristics stratified by gender in Table 1.

Q2 also table 2 must be stratified by gender

Response: We also added results stratified by gender into Table S6-S7.

Q3 into the discussion please differentiate the level of sarcopenia with the condition of sarcopenic obesity citing this paper. Perna, S., Peroni, G., Faliva, M.A., Bartolo, A., Naso, M., Miccono, A. and Rondanelli, M., 2017. Sarcopenia and sarcopenic obesity in comparison: prevalence, metabolic profile, and key differences. A cross-sectional study in Italian hospitalized elderly. Aging clinical and experimental research, 29(6), pp.1249-1258.

Response: Thanks for your significant suggestion. We have added this part (Line 281-282, “Sarcopenia obesity might also influence cognitive function because of malnutrition.”) and cited this paper.

Reviewer 2 Report

This is a generally well written and concise report of an important study with a relatively clear result and with the limitations discussed.

However it is missing its Figure 1.

Also the report would benefit from some explanation of the statistical analysis in terms of the use of generalized estimating equations used to examine the association of sarcopenia and its components with cognitive function and BDRM score during the longitudinal study. Thus at baseline the relationships are clearly shown in table 1 having divided the cohort into those with or without sarcopenia. However in tables 2 and 3 which show beta coefficiants for the association of sarcopenia and its components with cognitive function or BDRM score based on analysis at baseline and at at least one of three follow up times.  What is not clear is whether the analysis shows the extent of falling cognitive function or increasing BDRM score at baseline and at each follow up for just  those with sarcopenia at baseline, or for all subjects who either started with or developed sarcopenia over time. This needs clarification.

Finally some of the references are not shown in brackets (lines 251-253)

Author Response

Q1 it is missing its Figure 1.

Response: Thanks for your significant notes, we have added Fig 1 into the manuscript.

Q2 Also the report would benefit from some explanation of the statistical analysis in terms of the use of generalized estimating equations used to examine the association of sarcopenia and its components with cognitive function and BDRM score during the longitudinal study. Thus at baseline the relationships are clearly shown in table 1 having divided the cohort into those with or without sarcopenia. However in tables 2 and 3 which show beta coefficients for the association of sarcopenia and its components with cognitive function or BDRM score based on analysis at baseline and at least one of three follow up times.  What is not clear is whether the analysis shows the extent of falling cognitive function or increasing BDRM score at baseline and at each follow up for just those with sarcopenia at baseline, or for all subjects who either started with or developed sarcopenia over time. This needs clarification.

Response: Thanks for your comments. In this paper, we mimic a prospective study design and our analysis shows the extent of falling cognitive function or increasing BDRM score at each follow up for just those with sarcopenia at baseline.

Q3 Finally some of the references are not shown in brackets (lines 251-253)

Response: Thanks for your notes. We added brackets for the reference 9 (Line 271-272) and corrected others if needed.

Reviewer 3 Report

This study revealed the association between sarcopenia and cognitive function, and the topic is significantly interesting.

However, some modifications are needed as shown below.

- Line 84: Figure 1 did not appear in the text. You need to add it.

- Line 110 : What was the Sarcopenia measurement tool? DEXA or BIA? Although mentioned in the limitations of the study, it should be presented exactly with what equipment was measured.

- Line 251(7), 252(9), 253(20) + Reference: The reference cleanup does not seem to be correct. It should be presented properly.

- In addition, the thesis as a whole needs to be revised according to the style of the academic journal. It is less readable.

Author Response

Q1: Figure 1 did not appear in the text. You need to add it.

Response: Thanks for your notes. We have added Fig 1 into the manuscript.

Q2: What was the Sarcopenia measurement tool? DEXA or BIA? Although mentioned in the limitations of the study, it should be presented exactly with what equipment was measured.

Response: Thanks for your comments. As you mentioned, The CHARLS did not use bioelectrical impedance or X-ray to measure muscle mass, therefore we calculated muscle mass using ASM (an anthropometric equation in the Chinese population). We used handgrip strength to measure skeletal muscle strength and gait speed to measure physical performance provided by CHARLS. Sarcopenia was defined when low muscle mass plus low muscle strength or low physical performance were met. (Line 114-139)

Q3: The reference cleanup does not seem to be correct. It should be presented properly.

Response: Thanks for your notes, we have adjusted reference format and presented it properly.

Q4: In addition, the thesis as a whole needs to be revised according to the style of the academic journal. It is less readable.

Response: Thanks for your comments. We revised the manuscript according to the style of the academic journal and also invited an English native speaker to improve the readability.